# Longitudinal monitoring of *KRAS*-mutated circulating tumor DNA enables the prediction of prognosis and therapeutic responses in patients with pancreatic cancer

Fumiaki Watanabe[1], Koichi Suzuki[1]*, Sawako Tamaki[1], Iku Abe[1], Yuhei Endo[1], Yuji Takayama[1], Hideki Ishikawa[1], Nao Kakizawa[1], Masaaki Saito[1], Kazushige Futsuhara[1], Hiroshi Noda[1], Fumio Konishi[2], Toshiki Rikiyama[1]

1 Department of Surgery, Saitama Medical Center, Jichi Medical University, Amanuma-cho, Omiya-ku, Saitama, Japan, 2 Nerima Hikarigaoka Hospital, Hikarigaoka, Nerima-ku, Tokyo, Japan

* ksuzubnhm@yahoo.co.jp

## Abstract

### Background

Liquid biopsies enable the detection of circulating tumor DNA (ctDNA). However, the clinical significance of *KRAS*-mutated ctDNA for pancreatic cancer has been inconsistent with respect to its prognostic and predictive potential.

### Methods and findings

A total of 422 blood samples were collected from 78 patients undergoing treatments for localized and metastatic pancreatic ductal adenocarcinoma. *KRAS* mutation in tissues and *KRAS* ctDNA levels in plasma were determined by RASKET and droplet digital polymerase chain reaction. Longitudinal monitoring of *KRAS* ctDNA was performed to assess its significance for predicting recurrence and prognosis and for evaluating therapeutic responses to chemotherapy compared with carbohydrate antigen 19–9 (CA19-9). In 67 tumor tissues, discrepancies in point mutations of *KRAS* were rarely observed among individual patients, implying that one targeted point mutation of *KRAS* can be determined in tumor tissues prior to longitudinal blood monitoring. One-time blood assessment of *KRAS*-mutated ctDNA before surgery or chemotherapy was not clearly associated with recurrence and prognosis. Sequential blood monitoring was performed in 39 patients who underwent surgery for potentially resectable tumors. Increased CA19-9 levels were significantly associated with recurrence, but not prognosis ($P<0.001$, $P = 1.0$, respectively), whereas emergence of *KRAS* ctDNA was significantly associated with prognosis ($P<0.001$) regardless of recurrence. Furthermore, in 39 patients who did not undergo surgery, detection of *KRAS* ctDNA was a predictive factor for prognosis ($P = 0.005$). Multivariate analysis revealed that detection of *KRAS* ctDNA was the only independent prognostic factor regardless of tumor resection (hazard ratios = 54.5 for patients who underwent surgery and 10.1 for patients who did not undergo surgery; $P<0.001$ for both). Patients without emergence of *KRAS* ctDNA within 1

**Funding:** The present study was supported by Grant-in-Aid for Scientific Research (grant number JP 16K10514) from the Ministry of Education, Culture, Sports, Science and Technology and the JKA Foundation through its promotion funds from the Keirin Race (grant number 27-1-068 (2)). The funders had no role in study design, data collection and analysis, decision to publish, or preparation of the manuscript.

**Competing interests:** The authors have declared that no competing interests exist.

**Abbreviations:** AJCC, American Joint Committee on Cancer; CA19-9, carbohydrate antigen 19–9; ctDNA, circulating tumor DNA; ddPCR, droplet digital polymerase chain reaction; FFPE, formalin-fixed paraffin-embedded; *KRAS*, Kirsten rat sarcoma viral oncogene homolog; OS, overall survival; PDAC, pancreatic ductal adenocarcinoma; PFS, progression-free survival; RECIST, Response Evaluation Criteria in Solid Tumors; RFS, recurrence-free survival.

year after surgery showed significantly better prognosis irrespective of recurrence (*P*<0.001). No detection or disappearance of *KRAS* ctDNA within 6 months of treatment was significantly correlated with therapeutic responses to first-line chemotherapy (*P*<0.001). Changes in *KRAS* status provided critical information for the prediction of therapeutic responses.

## Conclusions

Our study showed for the first time that detection of *KRAS* ctDNA levels within a short period enables the prediction of prognosis and therapeutic responses in patients with pancreatic cancer.

## Introduction

Pancreatic ductal adenocarcinoma (PDAC) ranks as the fourth leading cause of cancer-related mortality in the United States and Japan [1, 2]. Surgical resection is considered the only curative treatment for PDAC. As PDAC is usually diagnosed at advanced stage, up to 20% of patients are suitable for initial resection [3]. Even after curative resection, most patients will experience recurrence within a year. The 5-year survival rate for patients undergoing complete resection is only approximately 25% [4]. Neoadjuvant chemotherapy to improve the treatment efficacy of surgery for patients with resectable tumors is under trial. The probability of complete resection may be lost even with a short period of neoadjuvant chemotherapy. Treatment without surgery results in unsatisfactory outcomes and poor prognosis with a median survival of 5–9 months [4], as observed in patients with unresectable tumors; nevertheless, unnecessary surgical treatment should be avoided.

Recent improvements in chemotherapy for patients with unresectable tumors have shown to prolong survival. The most effective treatment should be determined according to survival benefit for each patient; hence, an ideal predictive biomarker is required. For this purpose, carbohydrate antigen 19–9 (CA19-9) has been commonly used to establish diagnosis, assess resectability, monitor progression, and determine prognosis [5]. Despite the acceptance of the utility of CA19-9 as a valuable predictor for prognosis of PDAC, controversy remains as to the sensitivity and cutoff value of CA19-9.

As an alternative to CA19-9, detection of circulating cell-free DNA in the bloodstream, known as liquid biopsy, has been considered a predictive biomarker for invasive cancers [6–10]. Cell-free DNA originates from somatic DNA that is discharged into the systemic circulation following cellular necrosis and apoptosis [11]. Droplet digital polymerase chain reaction (ddPCR) and BEAMing technology are two widely used platforms for liquid biopsy, which can discover mutant alleles in the bloodstream with a high sensitivity of 0.001–0.01% [12, 13]. The detection of mutated circulating tumor DNA (ctDNA) provides prognostic and predictive information on various cancers [14, 15]. Liquid biopsy is an ideal noninvasive tool that allows multiple tests over time and provides real-time data on changes in tumors. Furthermore, liquid biopsy enables longitudinal monitoring of mutated ctDNA. Monitoring real-time changes within the tumor reflects tumor dynamics [16]. Our previous study showed that longitudinal monitoring of mutated ctDNA indicated tumor dynamics in connection with various treatments for patients with colorectal cancer, which in-turn provided useful information for treatment determination [17]. *KRAS* mutations have been detected in 50% of colorectal cancers and 90% of PDAC [4, 18, 19] and the heterogeneity of *KRAS* mutations between primary

tumor and metastasis in individual patients with PDAC is rare [20, 21], suggesting that, with respect to prognosis, circulating mutant *KRAS* ctDNA in the blood is a good biomarker for detecting the presence of cancer cells. In 1999, Castells et al. reported the association between poor survival and presence of *KRAS* mutations in plasma from patients with PDAC [22]. Later, several studies reported the feasibility of detecting circulating mutant *KRAS* genes in the blood of patients with PDAC, as well as the prognostic relevance of these genes [23, 24]. However, the clinical significance of *KRAS*-mutated ctDNA in PDAC has been inconsistent with respect to its prognostic and predictive potential [25, 26]. This inconsistency could be induced by limited points of detection of circulating mutant *KRAS* genes in the blood. In the past, assessments were performed at a few time points, for instance, just before and after surgery. However, longitudinal monitoring has not been attempted.

In this study, we evaluated the significance of sequentially assessing *KRAS* ctDNA levels through longitudinal monitoring. Here, we demonstrated for the first time that detection of *KRAS* ctDNA levels within a short period enables the prediction of prognosis and therapeutic responses in patients with PDAC.

## Methods

### Patients and study design

We prospectively recruited 78 patients clinically diagnosed with localized, metastatic, and recurrent PDAC and collected 422 blood samples between June 2014 and December 2017 at Saitama Medical Center, Jichi Medical University, Japan. Schematic of patient recruitment and our study endpoints are shown in S1 Fig. Characteristics of the 39 patients who underwent surgery and 39 patients who did not are shown in S1 Table, respectively. This study is conducted as an exploratory study without calculation of sample size for primary endpoints. After surgery, signs of recurrence were confirmed based on imaging findings. At least three serial liquid biopsy samples were obtained postoperatively from patients who underwent surgery. Therapeutic response of tumors was assessed based on the Response Evaluation Criteria in Solid Tumors (RECIST) version 1.1 [27] for routine clinical evaluation. At least two serial liquid biopsy samples were obtained from patients who did not undergo surgery. However, four patients provided only one sample each because of their death. The median follow-up time for all patients was 16.2 months. All patients provided written informed consent for the examination of their tissue and plasma and the use of their clinical data. The study protocol was approved by the research ethics committee of Jichi Medical University and conformed to the ethical guidelines of the World Medical Association Declaration of Helsinki.

### Analysis of *KRAS* status in PDAC tissues

*KRAS* status in PDAC tissues was evaluated by RASKET with a sensitivity of 1–5% and ddPCR with a sensitivity of 0.01–0.1%, using formalin-fixed paraffin-embedded (FFPE) tumor tissues, including endoscopic ultrasound-guided fine-needle aspiration samples. *KRAS* status in 67 tumor tissues was analyzed using RASKET by a clinical testing company (Special Reference Laboratories, Tokyo, Japan). Subsequently, tissue DNA was extracted from 73 FFPE tissues using QIAamp DNA FFPE Tissue Kit (Qiagen, Hilden, Germany) according to the manufacturer's instructions. Early reports have shown that point mutations at codon 12 of *KRAS* oncogene mostly include G12V, G12D, and G12R, whereas other types of point mutations of *KRAS* are rarely detected in patients with PDAC [19, 28, 29]. Therefore, these three types of *KRAS* mutations were predominantly identified by ddPCR. In addition, Q61H, another type of *KRAS* mutation that emerged prior to drug resistance was verified in four patients by ddPCR

after initial determination by RASKET. *KRAS* status in five patients could not be assessed because of insufficient DNA samples.

## Analysis of heterogeneity of *KRAS* mutations in PDAC by ddPCR

Slices with a thickness of 10 μm were obtained for each FFPE tissue specimen from patients who underwent surgery. Deparaffinization, rehydration, and hematoxylin and eosin staining were performed under enzyme-free conditions. The slides were subsequently placed on poly-ethylene naphthalate membrane slides (Leica Microsystems, Wetzlar, Germany) for laser microdissection using LMD 7000 (Leica Microsystems, Wetzlar, Germany), and the tumor center and invasion front were then isolated from each slide (S2 Fig). After laser microdissection, DNA was extracted using the QIAamp DNA FFPE Tissue Kit (Qiagen, Hilden, Germany) according to the manufacturer's instructions. *KRAS* mutations were assessed using ddPCR; available metastatic or recurrent tumors (N = 13) were also examined.

## Plasma sample collection and processing

A total of 422 blood samples were collected from patients with localized, metastatic, and recurrent PDAC at Saitama Medical Center, Jichi Medical University. From each patient, 7 mL of whole blood was drawn into EDTA-containing tubes, and plasma was collected by centrifugation at $3000 \times g$ for 20 min at 4°C, followed by centrifugation at $16000 \times g$ for 10 min at 4°C in a fresh tube. Plasma samples were separated from peripheral blood cells and stored at -80°C until DNA extraction.

## Extraction of circulating cell-free DNA

Circulating cell-free DNA was extracted from 2 mL of plasma using QIAamp Circulating Nucleic Acid Kit (Qiagen, Hilden, Germany) according to the manufacturer's instructions.

## ddPCR analyses

*KRAS* status in tumor tissues and plasma was analyzed using the Bio-Rad QX200 ddPCR system (Bio-Rad Laboratories, Hercules, CA, USA). We used a commercially available PrimePCR *KRAS* kit for ddPCR. *KRAS* mutations in each blood sample were verified according to the corresponding mutation (C12V, G12D, G12R, and Q61H) in matched tumor tissues determined by ddPCR (S3 Fig). The reaction mixture comprised of 10 μL of 2× ddPCR Supermix, 1 μL of each reference and variant 20× Bio-Rad PrimePCR *KRAS* for ddPCR, and 10 μL of sample eluted from plasma in a final volume of 22 μL. The mixture was loaded onto a DG8 cartridge (Bio-Rad Laboratories, Hercules, CA, USA) with 70 μL of droplet generation oil, and the cartridge was placed into the droplet generator. The generated droplets (approximately 15000 generated droplets per well) were transferred to a 96-well reaction plate, heat-sealed with a foil lid, and subjected to thermocycling in a Veriti thermal cycler (Thermo Fisher Scientific, Waltham, MA, USA) under the following cycling conditions: 95°C for 10 min and 40 cycles at 95°C for 30 s and subsequently at 55°C for 90 s. Amplified droplets were analyzed using a QX200 droplet reader (Bio-Rad Laboratories, Hercules, CA, USA) for the fluorescent measurement of FAM and HEX probes for wild-type and mutant genes, respectively. Quanta-Soft software (Bio-Rad Laboratories, Hercules, CA, USA) was used to measure the number of positive and negative droplets, and their ratio (mutated allele frequency) was fitted to a Poisson distribution to determine the target copy number/mL in the input reaction. Samples with two or more positive droplets were considered positive according to threshold values, which we

previously reported [17]. For data reproducibility, analysis of *KRAS* status in tumor tissues and plasma was performed in duplicate or triplicate.

## Statistical analysis

To assess prognosis, we measured recurrence-free survival (RFS), progression-free survival (PFS), and overall survival (OS) as our endpoints. RFS was defined as the time from surgery to confirmation of recurrence based on radiological findings. PFS was defined as the time from pathological diagnosis to disease progression according to RECIST 1.1 or cancer-related mortality in treatment-naïve patients receiving chemotherapy. OS in patients who underwent surgery was defined by the time from surgery to occurrence of the event, whereas OS in patients who did not undergo surgery was defined by the time from pathological diagnosis to the occurrence of event. A Cox proportional hazards regression model was used to evaluate the association between overall mortality and other factors in univariate and multivariate analyses. The following variables were analyzed in patients who underwent surgery: sex; age at surgery (≤70 years versus >70 years); neoadjuvant chemotherapy (yes versus no); surgical methods (subtotal stomach-preserving pancreaticoduodenectomy and total pancreatectomy versus distal pancreatectomy); tumor size (≤2 cm versus >2 cm); pathological differentiation (well versus moderate and others); American Joint Committee on Cancer (AJCC) T factor (T1+T2 versus T3+T4); lymph node metastasis (negative versus positive); AJCC stage (IA/IB/IIA versus IIB/III/IV); preoperative CA19-9 level (≥37 U/mL versus <37 U/mL); presence of *KRAS*-mutated ctDNA before surgery (negative versus positive); and emergence of *KRAS*-mutated ctDNA and CA19-9 levels upon longitudinal monitoring. Additionally, the following variables were analyzed in patients who did not undergo surgery: sex; age (≤70 years versus >70 years); AJCC stage (III versus IV); CA19-9 level (≥37 U/mL versus <37 U/mL); presence of *KRAS*-mutated ctDNA (negative versus positive). RFS, PFS and OS curves were constructed using the Kaplan-Meier method. The emergence of *KRAS*-mutated ctDNA and CA19-9 levels were regarded as time-dependent covariates in longitudinal monitoring. To graphically illustrate the effects of *KRAS*-mutated ctDNA and CA19-9 on OS, Simon–Makuch plots were generated with a landmark placed at the median point of detection of *KRAS*-mutated ctDNA and CA19-9 >37 U/mL in patients who did and did not undergo surgery. Several factors with a *P*-value of <0.15 in univariate analysis were subjected to multivariate analysis. A *P*-value of 0.05 was considered statistically significant. Fisher's exact test was used for categorical variables such as the presence of *KRAS*-mutated ctDNA, CA19-9 level (≥37 U/mL versus <37 U/mL), and outcome (dead or alive). All statistical analyses were performed using EZR version 1.31 (Saitama Medical Center, Jichi Medical University, Saitama, Japan). We also used R version 3.1.1 (The R Foundation for Statistical Computing, Vienna, Austria) for graphical interface.

## Results

### *KRAS* assessment in tumor tissue samples and investigation of heterogeneity

In advance of the investigation of *KRAS*-mutated ctDNA in plasma, *KRAS* assessment was performed in tumor tissues of 67 patients with PDAC using RASKET with a sensitivity of 1–5% and ddPCR with a sensitivity of 0.01–0.1%. With respect to frequency, G12V, G12D, D12R, Q61H, G12V+G12R, and G12D+G12R and wild-type *KRAS* alleles were detected in 24 (35.8%), 22 (32.8%), 6 (9.0%), 4 (6.0%), 2 (3.0%), 1 (1.5%), and 8 (11.9%) out of 67 samples, respectively. Of 67 patients, 59 (88.1%) with *KRAS* mutation (assigned from No. 1 to No. 59) and 8 (11.9%) patients without (assigned from No. 60 to No. 67) were identified by *KRAS*

assessment using RASKET (S3 Fig). These 8 patients without *KRAS* mutation detected by RASKET showed the presence of *KRAS* mutations by ddPCR. *KRAS* mutations in these 8 patients were present at less than 1%, but ddPCR identified these rare mutations because of its high sensitivity. When the *KRAS*-mutated allele frequency was ≥1% based on ddPCR, the *KRAS* mutations detected by RASKET were identical to those by ddPCR; as observed in the 59 patients whose mutations identified by RASKET corresponded to those determined by ddPCR (S3 Fig).

Moreover, we investigated the heterogeneity of *KRAS* mutations between tumor center and invasion front, as well as between primary tumor and metastasis, which revealed that discrepancies in *KRAS* status among individual patients were rarely observed. For *KRAS* mutations with frequency ≥1%, the concordance between tumor center and invasion front and between primary tumor and metastasis was 94.7% and 90.9%, respectively, implying the absence of heterogeneity in *KRAS* mutations (S4 Fig). Thus, point mutations identified in tissues should be monitored and detected in blood with no additional exploration required.

### One-point assessment of *KRAS* ctDNA and CA19-9 levels before surgery was not associated with RFS, but *KRAS* ctDNA before chemotherapy was a potential predictive prognostic factor, whereas CA19-9 prior to chemotherapy was not

We evaluated the significance of one-point assessment in *KRAS* ctDNA and CA19-9 levels before treatments, surgery and chemotherapy, to predict treatment outcome. Among 39 patients who underwent surgery, *KRAS*-mutated ctDNA was detected in 7 patients, and 28 patients had CA19-9 >37 U/mL. There was no significant difference in RFS by neither *KRAS*-mutated ctDNA status nor CA19-9 level ($P = 0.38$, $P = 0.7$, respectively; Fig 1A). No effect of the presence of *KRAS*-mutated ctDNA or CA19-9 level before surgery on RFS (median: 16.9 versus 32.4 months and 16.9 versus 17.1 months, respectively; Fig 1A) was observed. With respect to one-point assessment before chemotherapy, 26 chemotherapy-naïve patients were assessed before first-line chemotherapy among 39 patients who did not undergo surgery. *KRAS*-mutated ctDNA was detected in 12 patients, and 20 patients had CA19-9 >37 U/mL. Although there was no significant difference in prognosis by neither *KRAS*-mutated ctDNA status nor CA19-9 level ($P = 0.07$, $P = 0.86$, respectively; Fig 1B), the presence of *KRAS*-mutated ctDNA before chemotherapy was a potential predictive prognostic factor. The median OS of patients with and without detection of *KRAS*-mutated ctDNA was 15.8 and 33.7 months, respectively, whereas the median OS of patients with CA19-9 level ≥37 U/mL and <37 U/mL was 16.6 and 19.8 months, respectively (Fig 1B).

### Sequential assessments of ctDNA and CA19-9 in longitudinal monitoring to assess treatment outcome

Fig 2 shows sequential assessments of ctDNA and CA19-9 in longitudinal tests with respect to recurrence and prognosis. Fig 2A presents a comparison of *KRAS*-mutated ctDNA and CA19-9 in longitudinal tests for 39 patients who underwent surgery. Twenty-two patients showed recurrence and Fisher's exact test indicated that increased CA19-9 levels were significantly associated with recurrence ($P<0.001$). However, increased CA19-9 levels were not linked to prognosis ($P = 1.0$). On the other hand, emergence of *KRAS* ctDNA in longitudinal tests was associated with prognosis ($P<0.001$) regardless of recurrence, which emphasized its significance as a prognostic factor. Fig 2B presents a comparison of *KRAS*-mutated ctDNA and CA19-9 in longitudinal evaluation of 39 patients who did not undergo surgery. Fisher's exact test indicated that detection of *KRAS*-mutated ctDNA in longitudinal tests was associated with

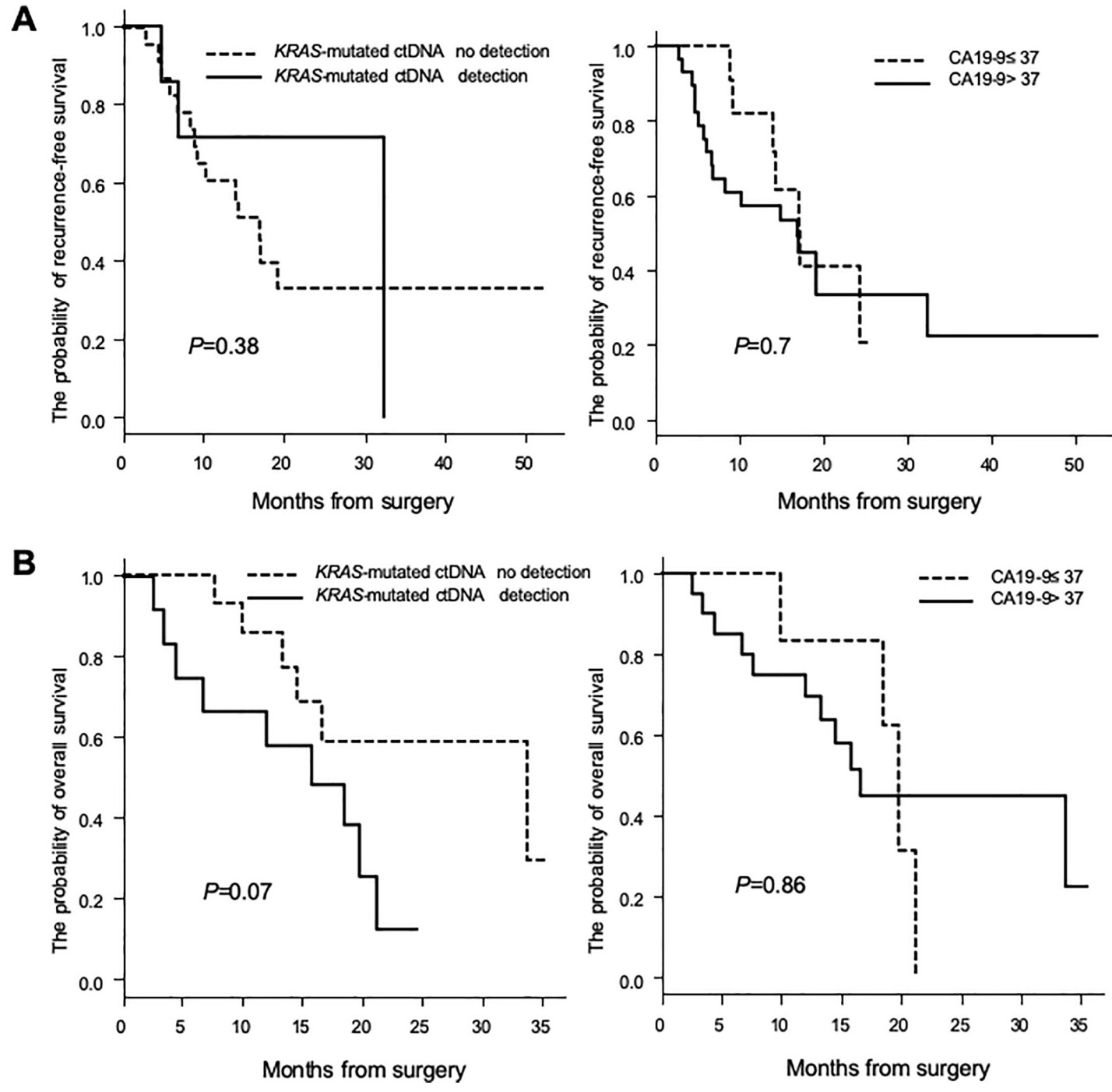

**Fig 1. One-point assessment of ctDNA and carbohydrate antigen 19–9 (CA19-9) before treatment to assess recurrence and prognosis.** (A) Recurrence-free survival (RFS) curves according to the presence of *KRAS*-mutated ctDNA and CA19-9 level before surgery in 39 patients who underwent surgery ($P$ = 0.38 and 0.7 by log-rank test). (B) Overall survival (OS) curves according to the presence of *KRAS*-mutated ctDNA and CA19-9 level before first-line chemotherapy in 26 chemotherapy-naïve patients who did not undergo surgery ($P$ = 0.07 and 0.86 by log-rank test). X-axes indicate the months from surgery, whereas Y-axes indicate the probability of RFS or OS.

prognosis ($P$ = 0.005), whereas CA19-9 was not ($P$ = 0.692). Details on the clinical course of 39 patients who underwent surgery and 39 patients who did not are shown in S2 and S3 Tables,

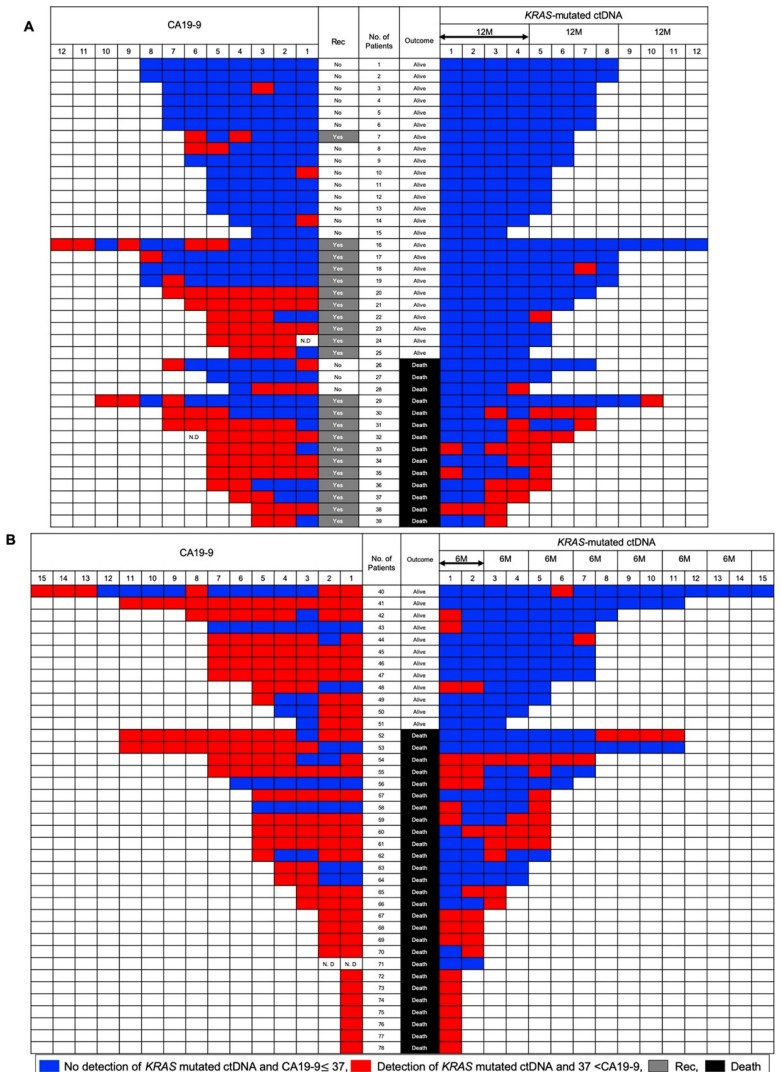

**Fig 2. Sequential assessments of *KRAS*-mutated ctDNA and carbohydrate antigen 19–9 (CA19-9) level in longitudinal tests.** (A) CA19-9 levels and the emergence of *KRAS*-mutated ctDNA are shown under "CA19-9" and "*KRAS*-mutated ctDNA," respectively, and are ordered as per the timing of blood examination after surgery (1→12). CA19-9 ≤37 U/mL and no detection of *KRAS*-mutated ctDNA are represented in blue, whereas CA19-9 >37 U/mL and the emergence of *KRAS*-mutated ctDNA are represented in red. Recurrence is shown under "Rec," with "no" and "yes" indicated in white and gray, respectively. Prognosis is shown under "Outcome," with "alive" and "death" indicated in white and gray, respectively. Examination results for every 3 months are shown in one cell; thus, four cells correspond to approximately 1 year. ND, not determined. (B) Sequential assessments of *KRAS*-mutated ctDNA and CA19-9 levels in longitudinal tests for patients who did not undergo surgery. CA19-9 levels and the emergence of *KRAS*-mutated ctDNA are ordered as per the timing of blood examination (1→15). Number of patients corresponds to those described in S2 and S3 Tables.

respectively. To demonstrate the significance of *KRAS*-mutated ctDNA monitoring, we present two cases of patients (one who underwent surgery and one who did not) exhibiting tumor dynamics in S5 Fig.

## Univariate and multivariate analyses of OS

Table 1 presents the 13 independent demographic and clinicopathological variables used in univariate analysis for OS of patients who underwent surgery. Four variables—namely,

**Table 1. Univariate and multivariate analyses of overall survival in patients who underwent surgery.**

| | Univariate analysis | | Multivariate analysis | |
| --- | --- | --- | --- | --- |
| Prognostic factors | Hazard ratio (95% CI) | *P*-value | Hazard ratio (95% CI) | *P*-value |
| Sex | | | | |
| Male | 1 | Reference | | |
| Female | 0.76 (0.26–2.21) | 0.61 | | |
| Age at surgery (median, 69.5 years) | | | | |
| <70 years | 1 | Reference | | |
| ≥70 years | 1.31 (0.45–3.80) | 0.62 | | |
| Neoadjuvant chemotherapy | | | | |
| No | 1 | Reference | 1 | Reference |
| Yes | 3.10 (0.82–11.7) | 0.09 | 0.62 (0.13–2.85) | 0.53 |
| Operation methods | | | | |
| SSPPD and total pancreatectomy | 1 | Reference | | |
| Distal pancreatectomy | 0.72 (0.24–2.21) | 0.57 | | |
| Tumor size | | | | |
| ≤2 cm | 1 | Reference | | |
| >2 cm | 2.98 (0.66–13.5) | 0.16 | | |
| Pathological differentiation | | | | |
| Well | 1 | Reference | 1 | Reference |
| Moderate and others[a] | 4.12 (1.38–12.3) | 0.01 | 1.93 (0.57–6.49) | 0.29 |
| AJCC T factor | | | | |
| T1/T2 | 1 | Reference | | |
| T3/T4 | 3.05 (0.397–23.37) | 0.28 | | |
| Lymph node metastasis | | | | |
| Negative | 1 | Reference | | |
| Positive | 0.81 (0.28–2.33) | 0.69 | | |
| AJCC stage | | | | |
| IA/IB/IIA | 1 | Reference | | |
| IIB/III/IV | 0.81 (0.27–2.33) | 0.69 | | |
| Preoperative CA19-9 level | | | | |
| ≤37 U/mL | 1 | Reference | | |
| >37 U/mL | 0.99 (0.31–3.16) | 0.98 | | |
| Presence of ctDNA before surgery | | | | |
| Negative | 1 | Reference | | |
| Positive | 0.66 (0.14–3.0) | 0.58 | | |
| CA19-9 status in monitoring | 9.4 (1.23–72.2) | 0.03 | | |
| Emergence of ctDNA in monitoring | 57.2 (7.4–442.4) | <0.001 | 54.5 (6.64–447.6) | <0.001 |

[a]Others include poorly, scirrhous, and adenosquamous; CI, confidence interval; SSPPD, subtotal stomach-preserving pancreaticoduodenectomy; AJCC, American Joint Committee on Cancer; CA19-9, carbohydrate antigen 19–9; ctDNA, circulating tumor DNA.

neoadjuvant chemotherapy, pathological differentiation, CA19-9 levels, and emergence of *KRAS*-mutated ctDNA were identified as prognostic factors. Multivariate Cox proportional hazards regression model indicated that the emergence of *KRAS*-mutated ctDNA (hazard ratio = 54.5, confidence interval: 6.64–447.6, *P*<0.001) was a significant factor for survival in patients who underwent surgery (Table 1). Considering the small number in this analysis, multivariate analysis for two time-dependent factors was not feasible; hence, CA19-9 was excluded from the analysis. To graphically illustrate the effects of *KRAS*-mutated ctDNA and CA19-9 in

longitudinal analyses on OS, Simon–Makuch plots were constructed with a landmark placed at the median time point (approximately 8 months and approximately 4 months, respectively) of the emergence of *KRAS*-mutated ctDNA and CA19-9 >37 U/mL (Fig 3A and 3B). Table 2 presents 6 independent demographic and clinicopathological variables used in univariate analysis for OS of patients who did not undergo surgery. In univariate analysis, the emergence of *KRAS*-mutated ctDNA was identified as one of the prognostic factors. Multivariate Cox proportional hazards regression model indicated that the emergence of *KRAS*-mutated ctDNA (hazard ratio = 10.4, confidence interval: 2.95–37.0, $P<0.001$) was the only significant factor for survival in patients who did not undergo surgery (Table 2). To graphically illustrate the effects of *KRAS*-mutated ctDNA and CA19-9 on OS, Simon–Makuch plots were generated with a landmark placed at the median time point (approximately 2 months) of the detection of *KRAS*-mutated ctDNA in patients who did not undergo surgery (Fig 3C). CA19-9 was not used to construct a survival curve with a time-dependent covariate, because most initial values of CA19-9 were over 37 U/mL.

## Sequential assessments of ctDNA within 1 year to assess prognosis in patients who underwent surgery

As presented in Fig 2A, patients demonstrating emergence of ctDNA within 1 year after surgery showed poor prognosis regardless of recurrence after surgery; hence, we re-evaluated the outcome by comparing patients with emergence of ctDNA within 1 year to patients without emergence. A statistically significant difference in OS according to *KRAS* status in blood was observed ($P<0.001$; Fig 4A). The emergence of *KRAS*-mutated ctDNA within 1 year after surgery was significantly associated with worse OS (median: not applicable versus 13.4 months).

## Sequential assessments of ctDNA within 6 months to assess therapeutic responses of chemotherapy-naïve patients

To evaluate therapeutic responses, we recruited 26 chemotherapy-naïve patients. Drug response was assessed using RECIST within 6 months of chemotherapy. A statistically significant difference in PFS was observed between patients in whom *KRAS*-mutated ctDNA was detected and those in whom *KRAS*-mutated ctDNA was not detected or disappeared within 6 months of chemotherapy ($P<0.001$; Fig 4B). The emergence of *KRAS*-mutated ctDNA within 6 months of chemotherapy was significantly associated with worse PFS (median: 14.9 versus 4.8 months). Changes in *KRAS* status provided critical information for the prediction of therapeutic responses. However, an analysis of patients according to CA19-9 levels to construct a survival curve for assessing therapeutic responses was not performed because CA19-9 did not show any change during treatments.

## Discussion

Our study showed the significance of sequential *KRAS* ctDNA assessments for predicting prognosis and therapeutic responses in patients with PDAC via longitudinal monitoring. In contrast, one-time assessment of *KRAS*-mutated ctDNA before surgery or chemotherapy was not clearly associated with recurrence and prognosis. Longitudinal monitoring of *KRAS*-mutated ctDNA enabled us to inform predictive significance within a short period after initial monitoring. Patients without emergence of *KRAS* ctDNA within 1 year after surgery showed significantly better prognosis irrespective of recurrence ($P<0.001$). No detection or disappearance of *KRAS* ctDNA within 6 months of treatment was significantly correlated with therapeutic responses to first-line chemotherapy ($P<0.001$). Changes in *KRAS* status provided critical

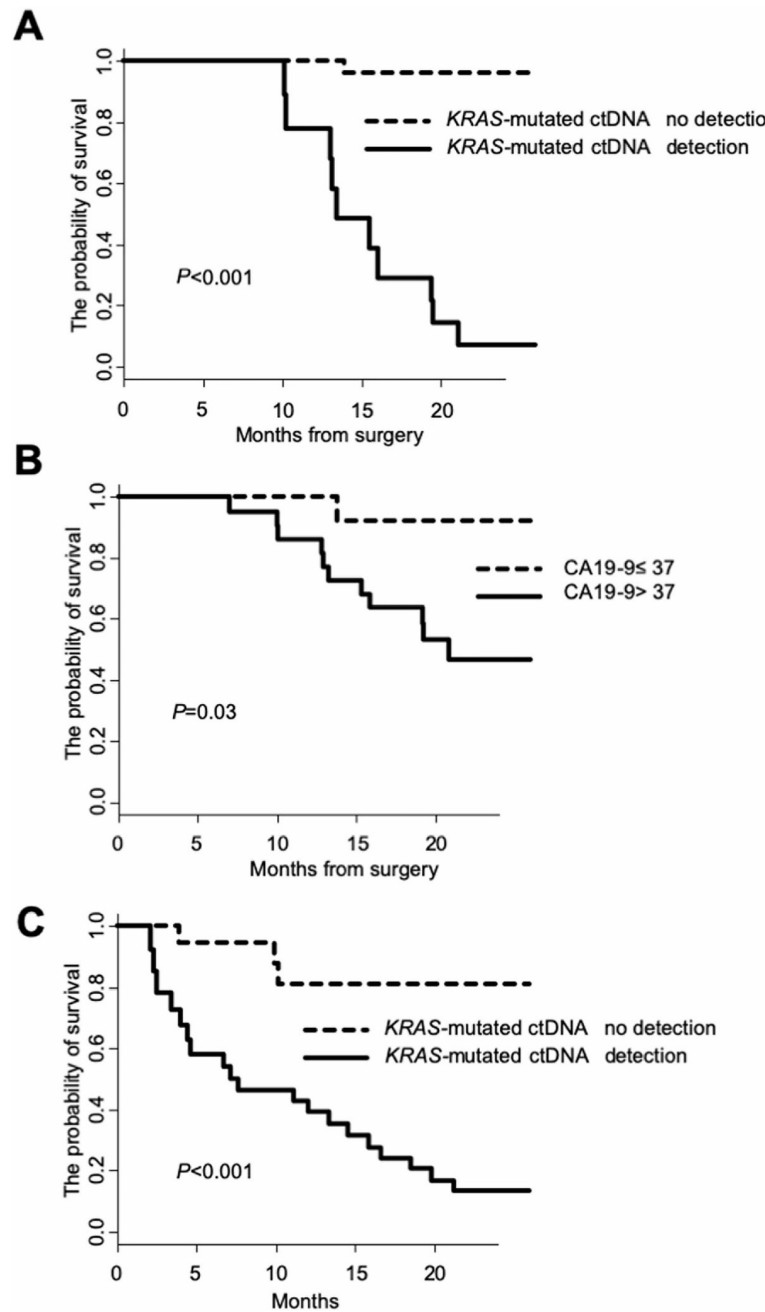

**Fig 3. Outcome according to *KRAS*-mutated ctDNA and CA19-9 in longitudinal evaluations.** (A) Simon–Makuch plot for the effect of emergence of *KRAS*-mutated ctDNA on overall survival (OS) in patients who underwent surgery, illustrated by a landmark at 8 months, the median time point for the detection of *KRAS*-mutated ctDNA (P<0.001). (B) Simon–Makuch plot for the effect of CA19-9 level on OS in patients who underwent surgery, illustrated by a landmark at 4 months, the median time point for CA19-9 level increasing to >37 U/mL (P = 0.03). (C) Simon–Makuch plot for the effect of emergence of *KRAS*-mutated ctDNA on OS in patients who did not undergo surgery, illustrated by a landmark at 2 months, the median time point for the detection of *KRAS*-mutated ctDNA (P<0.001). X-axes in Fig 3A and 3B indicate the months from surgery, X-axis in Fig 3C indicates the months from the initial evaluation in this study, whereas Y-axes indicate the probability of survival.

information towards prediction of therapeutic responses. Our study showed for the first time that assessment of *KRAS*-mutated ctDNA within a short period enables the prediction of prognosis and therapeutic responses in patients with PDAC.

**Table 2. Univariate and multivariate analyses of overall survival in patients who did not undergo surgery.**

| | Univariate analysis | | Multivariate analysis | |
|---|---|---|---|---|
| Prognostic factors | Hazard ratio (95% CI) | *P*-value | Hazard ratio (95% CI) | *P*-value |
| Sex | | | | |
| Male | 1 | Reference | | |
| Female | 0.82 (0.38–1.78) | 0.61 | | |
| Age (median, 69.5 years) | | | | |
| <70 years | 1 | Reference | | |
| ≥70 years | 1.73 (0.80–3.71) | 0.16 | | |
| AJCC stage | | | | |
| Stage III | 1 | Reference | | |
| Stage IV | 1.27 (0.55–2.92) | 0.58 | | |
| CA19-9 level[a] | | | | |
| ≤37 U/mL | 1 | Reference | | |
| >37 U/mL | 1.02 (0.45–2.28) | 0.97 | | |
| Presence of ctDNA[a] | | | | |
| Negative | 1 | Reference | | |
| Positive | 2.12 (0.99–4.57) | 0.05 | 0.55 (0.22–1.39) | 0.21 |
| Emergence of ctDNA in monitoring | 6.75 (2.23–19.9) | <0.001 | 10.4 (2.95–37.0) | <0.001 |

[a]Initial evaluation in monitoring; CI, confidence interval; AJCC, American Joint Committee on Cancer; CA19-9, carbohydrate antigen 19–9; ctDNA, circulating tumor DNA.

With respect to heterogeneity, molecular heterogeneity within the primary tumor (intratumoral heterogeneity) and between the primary tumor and metastatic lesions has been described previously [30–33] and is associated with resistance to various treatments [34]. Some studies have reported discrepancies between the primary tumor and *KRAS*-mutated ctDNA [6, 35, 36]. Hashimoto et al. [20] reported that intratumoral heterogeneity was

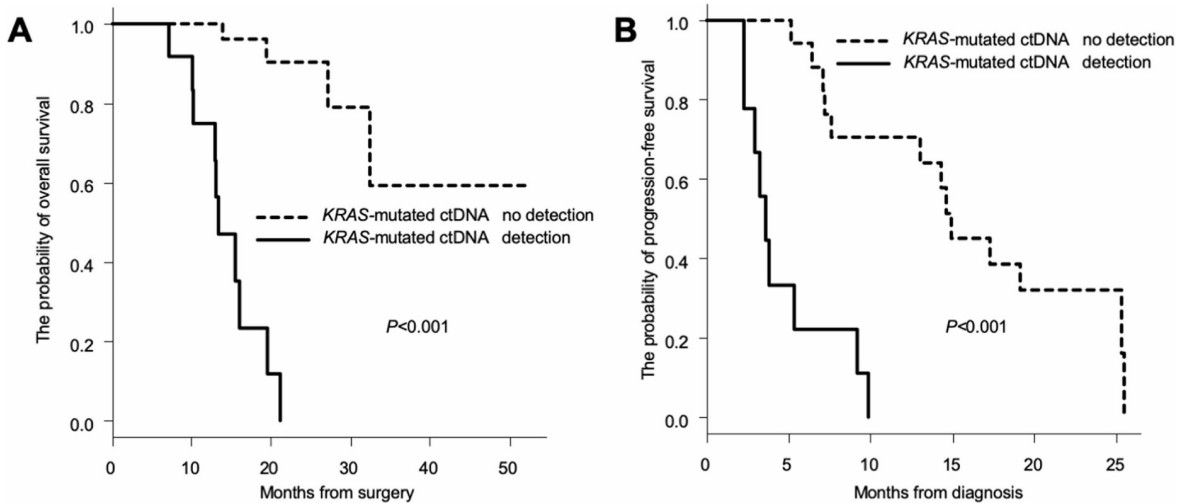

**Fig 4. Sequential assessments of *KRAS*-mutated ctDNA within a short period to assess outcome.** (A) Overall survival (OS) curves according to the emergence of *KRAS*-mutated ctDNA within 1 year after surgery (*P*<0.001 by log-rank test). (B) Progression-free survival (PFS) curves according to the emergence of *KRAS*-mutated ctDNA within 6 months of chemotherapy (*P*<0.001 by log-rank test). X-axes indicate the months from surgery and diagnosis, whereas Y-axes indicate the probability of OS and PFS.

observed in *KRAS* mutations between tumor centers and invasion fronts in 4.1% patients with pancreatic cancer. Makohon-Moore et al. [21] detected identical mutations in driver genes such as *KRAS* and *SMAD4* and reported no discrepancy in driver genes between primary tumors and distant metastases. In context of this controversy, we investigated the heterogeneity of *KRAS* mutations in PDAC in advance. The concordance between tumor center and invasion front and between primary tumor and metastasis was more than 90.0%; thus, the types of mutations detected in tissues were investigated by ddPCR in the blood. Discrepancies in point mutations of *KRAS* were rarely observed among individual patients, implying that a targeted point mutation of *KRAS* can be determined in tumor tissues prior to longitudinal monitoring of blood.

In our study, the presence of *KRAS*-mutated ctDNA before surgery was not clearly associated with recurrence. No report has addressed the significant effect of *KRAS*-mutated ctDNA on RFS, suggesting that one-point assessment of *KRAS*-mutated ctDNA before surgery was not useful for predicting recurrence. Nonetheless, with respect to prognosis prediction, Hadano et al. reported that one-point assessment of ctDNA before surgery was associated with patient outcome [37]. In patients who did not undergo surgery, *KRAS*-mutated ctDNA was likely associated with prognosis. Two studies supported our data [25, 26], but they could not demonstrate the significance of *KRAS*-mutated ctDNA for prognosis prediction. In contrast, preoperative CA19-9 levels showed no association with RFS, and CA19-9 levels in patients who did not undergo surgery were not associated with OS. Different CA19-9 thresholds were reported to predict RFS or OS [38, 39]; therefore, the optimal cut-off value of CA19-9 needs further investigation.

Longitudinal monitoring of *KRAS*-mutated ctDNA is a great advantage of liquid biopsy. In our study, multiple testing over time enabled the evaluation of association between the presence of *KRAS*-mutated ctDNA and prognosis of PDAC irrespective of tumor resection. A literature search for relevant studies using the search terms "circulating DNA" and "pancreatic cancer" was conducted in PubMed, and 18 studies investigating *KRAS*-mutated ctDNA in patients with PDAC were identified. In their review, Gall et al. [40] reported a similar number of studies describing *KRAS*-mutated ctDNA in patients with PDAC. Thus far, studies on this subject remain few. Furthermore, except for our study and those of Bernard et al. [41] and Sausen et al. [42], few have addressed the importance of longitudinal monitoring for predicting the outcome of PDAC. Bernard et al. reported the usefulness of patients' exosome DNA in addition to *KRAS*-mutated ctDNA in PDAC. Considering the advantages of monitoring, they concluded that serial exosome DNA was significantly associated with eventual disease progression; in contrast, serial *KRAS*-mutated ctDNA was not significantly correlated with the presence or absence of progression. In our study, emergence of *KRAS*-mutated ctDNA was considered a time-dependent covariate in longitudinal monitoring, and its effect with respect to prognosis was shown for the first time. In addition, CA19-9 level was identified as a prognostic factor in patients who underwent surgery in univariate analysis; nonetheless, CA19-9 was not regarded as a time-dependent covariate because most baseline CA19-9 values were >37 U/mL.

We observed a change in *KRAS* status in blood of a patient who exhibited complete radiological response to chemotherapy. *KRAS*-mutated ctDNA disappeared in response to drug treatment. We evaluated therapeutic responses to chemotherapy by comparing patients in whom *KRAS*-mutated ctDNA was detected to patients in whom *KRAS*-mutated ctDNA was not detected or disappeared within 6 months of chemotherapy. The change in *KRAS* status in blood enabled us to demonstrate that *KRAS* status was associated with PFS. These results are congruent with those reported by Del Re et al. [25] and Kruger et al. [43] who showed that early changes in *KRAS*-mutated ctDNA levels were useful for monitoring treatment responses

in patients with PDAC. Moreover, longitudinal monitoring revealed the significance of *KRAS* mutation assessment within 1 year after surgery, which was strongly associated with outcome irrespective of recurrence. Patients in whom *KRAS*-mutated ctDNA was not detected within 1 year after surgery showed better prognosis and responded significantly better to chemotherapy even after recurrence. These findings suggest the significance of sequential assessment of *KRAS*-mutated ctDNA within a short period.

In conclusion, our study demonstrated for the first time that assessment of *KRAS*-mutated ctDNA using longitudinal evaluation enables the prediction of prognosis and therapeutic responses in patients with PDAC. Although our study results should be interpreted within the study limitations and further examinations are required to draw a definitive conclusion, we believe that our study casts greater light on the selection of patients with PDAC.

## Supporting information

**S1 Fig. Schematic of patient recruitment and our study endpoints.**
(TIF)

**S2 Fig. Representative tumor centers (TC) and invasion fronts (IF).**
(TIF)

**S3 Fig. Comparison of *KRAS* status in tumor tissues using RASKET and droplet digital polymerase chain reaction (ddPCR).** *KRAS* mutations by RASKET and ddPCR are indicated in red, *KRAS* mutations with frequencies <1% by ddPCR are displayed in pink, and wild types are presented in aqua. Blank indicates no detection of *KRAS* mutation. ND, not determined.
(TIF)

**S4 Fig. Comparison of *KRAS* mutations within the primary tumor (intratumoral heterogeneity) and between the primary tumor and metastatic lesions.** (A) Assessment of *KRAS* mutations between tumor center and invasion front using droplet digital polymerase chain reaction (ddPCR). *KRAS* mutations with frequencies ≥1% are indicated in red, whereas those with frequencies <1% are displayed in pink. Blank indicates no detection of *KRAS* mutation. ND, not determined. As for *KRAS* mutations with frequencies ≥1%, 36 tumors showed concordance between the tumor center and invasion front, accounting for 94.7%. Two tumors (no. 24 and no. 63) did not show concordance; the number of tumors corresponded to that presented in S3 Fig (B) *KRAS* mutations with frequencies ≥1% are indicated in red, whereas those with frequencies <1% are displayed in pink. Blank indicates no detection of *KRAS* mutation. ND, not determined; LN, lymph node; Local, local recurrence in residual pancreas. As for *KRAS* mutations with frequencies ≥1%, 10 tumors showed concordance between the primary tumor and metastasis, accounting for 90.9%. One tumor (no. 27) did not show concordance; the number of tumors corresponded to that presented in S3 Fig.
(TIF)

**S5 Fig. Clinical course of representative patients in longitudinal assessments.** (A) A patient underwent subtotal stomach-preserving pancreaticoduodenectomy for pancreatic head cancer and was treated with S1 as adjuvant chemotherapy. CT image is shown (a). *KRAS*-mutated ctDNA increased in advance of CA19-9, and potential lymph node recurrence was subsequently detected using CT imaging. Following identification of recurrence by CT imaging (b, white arrowhead), first-line treatment with nab-paclitaxel temporarily led to tumor shrinkage (c) but eventually resulted in death. X-axes indicate the days from surgery, whereas Y-axes indicate the CA19-9 value and mutated allele level. (B) A patient with pancreatic body cancer and para-aortic lymph node metastasis (d, yellow arrow and arrowhead), the pathological

diagnosis of which was established by fine-needle aspiration biopsy, was treated with FOLFIR-INOX as first-line chemotherapy. *KRAS*-mutated ctDNA decreased and disappeared after chemotherapy. However, 17 cycles of FOLFIRINOX led to peripheral neuropathy; thus, the regimen was discontinued and FOLFIRI was used for this patient. Positron emission tomography (PET) revealed an absence of accumulation in the pancreatic body and para-aortic lymph node (e, yellow arrow and arrowhead), suggesting complete response. X-axes indicate the days from surgery, whereas Y-axes indicate the CA19-9 value and mutated allele level. FOLFIRI-NOX, folinic acid, fluorouracil, irinotecan, and oxaliplatin; FOLFIRI, folinic acid, fluorouracil, and irinotecan.
(TIF)

**S1 Table.** A) Characteristics of patients who underwent surgery. B) Characteristics of patients who did not undergo surgery.
(DOC)

**S2 Table. Clinical information of patients who underwent surgery.**
(DOC)

**S3 Table. Clinical information of patients who did not undergo surgery.**
(DOC)

## Author Contributions

**Conceptualization:** Fumiaki Watanabe, Koichi Suzuki, Toshiki Rikiyama.

**Data curation:** Sawako Tamaki, Iku Abe, Yuhei Endo, Yuji Takayama, Hideki Ishikawa, Nao Kakizawa, Masaaki Saito, Kazushige Futsuhara, Hiroshi Noda.

**Investigation:** Fumiaki Watanabe.

**Methodology:** Fumiaki Watanabe.

**Writing – original draft:** Fumiaki Watanabe, Koichi Suzuki.

**Writing – review & editing:** Koichi Suzuki, Fumio Konishi, Toshiki Rikiyama.

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
