## [Decision Letter · Decision Letter 0]

22 Aug 2019

PONE-D-19-17596

Longitudinal monitoring of KRAS-mutated circulating tumor DNA enables the prediction of prognosis and therapeutic responses in patients with pancreatic cancer

PLOS ONE

Dear Dr. Suzuki,

Thank you for submitting your manuscript to PLOS ONE. After careful consideration, we feel that it has merit but does not fully meet PLOS ONE’s publication criteria as it currently stands. Therefore, we invite you to submit a revised version of the manuscript that addresses the points raised below by the reviewers during the review process.

We would appreciate receiving your revised manuscript by Oct 06 2019 11:59PM. To enhance the reproducibility of your results, we recommend that if applicable you deposit your laboratory protocols in protocols.io, where a protocol can be assigned its own identifier (DOI) such that it can be cited independently in the future. For instructions see: http://journals.plos.org/plosone/s/submission-guidelines#loc-laboratory-protocols

We look forward to receiving your revised manuscript.

Kind regards,

Surinder K. Batra

Academic Editor

PLOS ONE

**Journal Requirements:**

2. In the Methods, please provide a justification of the sample size used in this study.

3. Our staff editors have determined that your manuscript is likely within the scope of our Targeted Anticancer Therapies and Precision Medicine Call for Papers. This editorial initiative is headed by a team of Guest Editors for PLOS ONE: Andrew Cherniack, Anette Duensing, Steven Gray, Sunil Krishnan, Chandan Kumar-Sinha and Gayle Woloschak. The Collection will encompass a diverse range of research articles about the identification and classification of driver genes and somatic alterations, target and drug discovery, mechanisms of drug resistance, and early detection and screening.  Additional information can be found on our announcement page: https://collections.plos.org/s/targeted-anticancer-therapies.

If you would like your manuscript to be considered for this collection, please let us know in your cover letter and we will ensure that your paper is treated as if you were responding to this call. If you would prefer to remove your manuscript from collection consideration, please specify this in the cover letter.

4. Thank you for stating that “The funders had no role in study design, data collection and analysis, decision to publish, or preparation of the manuscript” in your financial disclosure.

Please also provide the name of the funders of this study (as well as grant numbers if available) in your financial disclosure statement.

3. Thank you stating the following in your competing interests statement:

"“The funders had no role in study design, data collection and analysis, decision to publish, or preparation of the manuscript”"

Please complete the competing interests section fully.  If NO authors have competing interests, please enter: "The authors have declared that no competing interests exist."

If Authors have competing interests please enter competing interest details beginning with this statement:

"I have read the journal's policy and the authors of this manuscript have the following competing interests: [insert competing interests here]"

**Comments to the Author**

1. Is the manuscript technically sound, and do the data support the conclusions?

Reviewer #1: Partly

Reviewer #2: Yes

2. Has the statistical analysis been performed appropriately and rigorously? 

Reviewer #1: Yes

Reviewer #2: Yes

3. Have the authors made all data underlying the findings in their manuscript fully available?

Reviewer #1: No

Reviewer #2: Yes

4. Is the manuscript presented in an intelligible fashion and written in standard English?

Reviewer #1: No

Reviewer #2: Yes

5. Review Comments to the Author

Reviewer #1: 1. The abstract submitted here is previously published in JCO by the same group and available online[DOI: 10.1200/JCO.2019.37.15_suppl.e15712 Journal of Clinical Oncology 37, no. 15_suppl]. therefore, to avoid self plagiarism, the abstract in the present manuscript need to be modified.

2. There are several repetition of sentences in the manuscript, e.g. [line 412-415] Hadano et al. reported that one-point assessment of ctDNA before surgery was associated with 412 patient outcome[37]. reported that one-point assessment of ctDNA before surgery was associated 413 with patient outcome. In patients who did not undergo surgery, KRAS-mutated ctDNA was likely 414 associated with prognosis.

Reviewer #2: Important and timely topic; however, the authors need to rewrite major portions of the discussion and overstate that they were the first group to report this finding: Bernard et al. Gastroenterology. 2019 January ; 156(1): 108–118.e4. doi:10.1053/j.gastro.2018.09.022. Several additional relevant papers are cited in this manuscript.

The Bernard paper along with many of the additional papers should be incorporated into the discussion so that the current study is put into proper context.

Minor comments.

Line 81.

Are you suggesting that delay in surgical resection while receiving neoadjuvant therapy was responsible for the tumor becoming unresectable? I think it is more likely that the patient was spared an unnecessary surgery since it is more likely that micrometastatic disease was present at time of diagnosis.

6. PLOS authors have the option to publish the peer review history of their article (what does this mean?). If published, this will include your full peer review and any attached files.

Reviewer #1: No

Reviewer #2: No

---

## [Author Response · Author response to Decision Letter 0]

18 Nov 2019

Responses to Reviewers’ Comments

General comments: 

The article entitled “longitudinal monitoring of KRAS-mutated circulating tumor DNA enables the prediction of prognosis and therapeutic responses in patients with pancreatic cancer” describes the significance of mutated KRAS ctDNA as a prognostic marker in the liquid biopsy samples of pancreatic cancer patients. I appreciate the efforts from the authors for performing the in-depth analysis. Because the similar work has been published previously with some of the closely matching conclusions, the submitted manuscript need to be redrafted before considering it for publication. In addition, there are major concerns related to manuscript writing, reference citation, and plagiarism. If considered for publication, following are my comments that need to be addressed:

Major comments:

1. Although the study has been performed on a bigger cohort of PC patients and include the longitudinal analysis of liquid biopsy samples, authors need to justify how the submitted manuscript is different than the previously published research on prognostic significance of mutKRAS ctDNA in PC patients. following are the references for similar kind of work: 

• Kruger, S., et al., Repeated mutKRAS ctDNA measurements represent a novel and promising tool for early response prediction and therapy monitoring in advanced pancreatic cancer. Ann Oncol, 2018. 29(12): p. 2348-2355.

We are in agreement the editor’s suggestion. Kruger et al. reported that a decrease in mutKRAS ctDNA levels during therapy was an early indicator of response to therapy during the first 4 weeks of treatment. In accordance with the reviewer’s suggestion, we have revised the manuscript as follows:

Discussion section, lines 404-406

“These results are congruent with those reported by Del Re et al. [25] and Kruger et al. [43] who showed that early changes in KRAS-mutated ctDNA levels were useful for monitoring treatment responses in patients with PDAC.”

• Cohen, J.D., et al., Combined circulating tumor DNA and protein biomarker-based liquid biopsy for the earlier detection of pancreatic cancers. Proceedings of the National Academy of Sciences, 2017. 114(38): p. 10202.

Cohen et al. aimed to detect the early stage of pancreatic cancer by combined circulating tumor DNA and protein biomarker-based liquid biopsy. The aim of their study is slightly different from the purpose of our study; hence, this paper was not cited in our manuscript.

• Tjensvoll, K., et al., Clinical relevance of circulating KRAS mutated DNA in plasma from patients with advanced pancreatic cancer. Molecular Oncology, 2016. 10(4): p. 635-643.

In the study by Tjensvoll et al., Kaplan–Meier survival analyses indicated that patients with positive ctDNA status before or after initiation of chemotherapy had shorter progression-free survival, albeit without statistical significance.

2. The abstract has been previously published in JCO and content has been copied in the submitted manuscript. It will be considered as self-plagiarism and therefore, authors are requested to modify the abstract [DOI: 10.1200/JCO.2019.37.15_suppl.e15712 Journal of Clinical Oncology 37, no. 15_suppl].

In view of your valuable comment, we have accordingly revised the abstract and modified it in a different form.

3. Authors focused on comparative analysis of mutKRAS ctDNA and CA19-9 in liquid biopsy samples ignoring their combined sensitivity in prediction of prognosis. Previous study by Cohen et al., 2017 in PNAS states that combination of ctDNA and protein-based markers increases the sensitivity.

Cohen et al. focused on detecting the early stage of pancreatic cancer by combined circulating tumor DNA and protein biomarker-based liquid biopsy. Our study aimed to evaluate the usefulness of KRAS ctDNA monitoring for the prediction of prognosis and therapeutic responses in patients with pancreatic cancer.

4. The references are neither updated nor cited appropriately, e.g. ref 01 is about pancreatic cancer statistics and it was published in 2009, although 2019 cancer statistics is available.

I apologize for the inappropriate reference. We have accordingly replaced the original reference with the reference #2 for the paper published in 2019.

5. The manuscript needs scientific writing with no repetition of sentences to maintain proper connectivity and flow of information. 

I apologize for the repetition of some sentences. In view of your comment, we have accordingly revised the manuscript.

6. Citations in the manuscript do not look appropriate as all the citations are placed after the period, which need to be corrected.

I apologize for this. As per your advice, we have corrected this matter in the revised manuscript, and all reference citations are now placed before the period or comma.

Specific comments:

1. Line 24-25; modify as “All of the other authors contributed to sample collection, data collection and interpretation, and manuscript review”.

In accordance with the reviewer’s suggestion, we have revised the manuscript as follows:

Title page, lines 24–25

“All of the other authors contributed to sample collection, data collection and interpretation, and manuscript review.”

2. Line 240 and 356; patients without……….need to be elaborated.

Based on the reviewer’s suggestion, we have revised the manuscript as follows:

Results section, lines 239–240

“These 8 patients without KRAS mutation detected by RASKET showed the presence of KRAS mutations by ddPCR.”

Results section, lines 325

“hence, we re-evaluated the outcome by comparing patients with emergence of ctDNA within 1 year to patients without emergence.”

3. Line 255-257; Highlighted in bold and sentence finished abruptly………….correction required.

In accordance with the reviewer’s suggestion, we have revised the manuscript as follows:

Results section, lines 253–255:

“One-point assessment of KRAS ctDNA and CA19-9 levels before surgery was not associated with RFS, but KRAS ctDNA before chemotherapy was a potential predictive prognostic factor, whereas CA19-9 prior to chemotherapy was not”

4. All figure legends are inserted in the main text. Please submit the separate file for figure legends.

I apologize for the inappropriate insertion of figure legends in the main text. Figure legends have now been placed at the end of the manuscript.

5. Line 364-365; remove the bold font and put a period at the end of the sentence.

As per the reviewer’s suggestion, we have revised the relevant text in the manuscript as follows:

Results section, lines 329–330:

“Sequential assessments of ctDNA within 6 months to assess therapeutic responses of chemotherapy-naïve patients”

6. Pancreatic cancer has been used multiple time and it could be abbreviated as PC.

In view of the reviewer’s comment, we have replaced the term “pancreatic cancer” in the text with the abbreviation “PDAC.”

7. Line 412-415; sentence has been repeated, correction needed. 

I apologize for the repeated sentence. We have accordingly deleted this in the manuscript as follows:

Discussion section, lines 373

“reported that one-point assessment of ctDNA before surgery was associated with patient outcome.”

8. Line 441-444; sentence has been repeated, correction needed. 

We did not identify any repeated sentence in lines 441–444.

9. There are relevant and recent references are missed by the authors therefore, references need to be updated. 

I apologize for the inappropriate references. In view of the reviewer’s comment, we have accordingly checked all references and appropriately updated them.

10. Figure 2; better resolution is required so that in print form, it should be readable.

In accordance with the reviewer’s advice, we have replaced the figure with Figure 2 with better resolution.

11. Supporting information; sentences are not ending with the period, written casually. Please revise the supporting information.

I apologize for this. We have accordingly revised the sentences in view of your comment.

12. Figure legends: Figure legends are not correctly written. X-and Y-axes are not defined in the legends and there is no statistical information mentioned in the legends. 

In accordance with the reviewer’s comment, we have revised the figure legends and defined the X- and Y-axes.

13. For data reproducibility, authors have not mentioned how many times the experiments were repeated to get the statistical significance. While writing the figure legends, please mention how many times the experiment were repeated with their statistical significance.

In view of the reviewer’s comment, we have added the following text in the Methods section:

Lines 194–195:

“For data reproducibility, analysis of KRAS status in tumor tissues and plasma was performed in duplicate or triplicate.”

---

## [Decision Letter · Decision Letter 1]

18 Dec 2019

Longitudinal monitoring of KRAS-mutated circulating tumor DNA enables the prediction of prognosis and therapeutic responses in patients with pancreatic cancer

PONE-D-19-17596R1

Dear Dr. Suzuki,

We are pleased to inform you that your manuscript has been judged scientifically suitable for publication and will be formally accepted for publication once it complies with all outstanding technical requirements.

With kind regards,

Surinder K. Batra

Academic Editor

PLOS ONE

Additional Editor Comments (optional):

Reviewers' comments:

Reviewer's Responses to Questions

**Comments to the Author**

1. If the authors have adequately addressed your comments raised in a previous round of review and you feel that this manuscript is now acceptable for publication, you may indicate that here to bypass the “Comments to the Author” section, enter your conflict of interest statement in the “Confidential to Editor” section, and submit your "Accept" recommendation.

Reviewer #1: All comments have been addressed

Reviewer #2: All comments have been addressed

2. Is the manuscript technically sound, and do the data support the conclusions?

Reviewer #1: Yes

Reviewer #2: Yes

3. Has the statistical analysis been performed appropriately and rigorously? 

Reviewer #1: Yes

Reviewer #2: Yes

4. Have the authors made all data underlying the findings in their manuscript fully available?

Reviewer #1: Yes

Reviewer #2: Yes

5. Is the manuscript presented in an intelligible fashion and written in standard English?

Reviewer #1: Yes

Reviewer #2: Yes

6. Review Comments to the Author

Reviewer #1: Dear Editor,

Authors have revised the manuscript and addressed all the concerns/comments raised during the review. My reviews for the revised manuscript are as following:

1. Authors have addressed most of the comments and highlighted the novelty of the work in the revised manuscript.

2. Reference citation and language of the manuscript is acceptable for publishing this work in PLOS One.

3. Authors could have modified abstract to a better extent to avoid repetition from their previous abstract publication in JCO.

4. Figures have been revised and submitted by authors with a better resolution and readable text/legend.

My recommendation to the editor is that manuscript should be considered for publication in PLOS One.

Reviewer #2: (No Response)

7. PLOS authors have the option to publish the peer review history of their article (what does this mean?). If published, this will include your full peer review and any attached files.

Reviewer #1: No

Reviewer #2: No

---

## [Editor Report · Acceptance letter]

20 Dec 2019

PONE-D-19-17596R1 

Longitudinal monitoring of KRAS-mutated circulating tumor DNA enables the prediction of prognosis and therapeutic responses in patients with pancreatic cancer 

Dear Dr. Suzuki:

I am pleased to inform you that your manuscript has been deemed suitable for publication in PLOS ONE. Congratulations! Your manuscript is now with our production department. 

With kind regards,

on behalf of

Prof. Surinder K. Batra 

Academic Editor

PLOS ONE